# Rapid Communication: Middle Pleistocene Transition as a Phenomenon of Orbitally Enabled Sensitivity to Initial Values

Mikhail Y. Verbitsky[1,2] and Anne Willem Omta[3]

[1]Gen5 Group, LLC, Newton, MA, USA
[2]UCLouvain, Earth and Life Institute, Louvain-la-Neuve, Belgium
[3]Department of Earth, Environmental, and Planetary Sciences, Case Western Reserve University, Cleveland, OH, USA
Correspondence: Mikhaïl Verbitsky (verbitskys@gmail.com)

**Abstract.** The Middle Pleistocene Transition (MPT), i.e., the "fast" transition from ~41- to ~100-kyr rhythmicity that occurred about 1 Myr ago, remains one of the most intriguing phenomena of the past climate. The cause of this period shift is generally thought to be a change within the Earth System, since the orbital insolation forcing does not change its pattern through the MPT. Using a dynamical model rooted in ocean chemistry, we advance several novel concepts here: (i) the MPT could be a dominant-period relaxation process that may be dependent on the initial state of the system, (ii) this sensitivity to the initial state is enabled by the orbital forcing, (iii) depending on the amplitude of the orbital forcing and initial values, the MPT could have been not just of the 40 – 80 kyr type, as we observe in the available data, but also of a 20 – 40, 40 – 120, or even 80 – 40 kyr type, (iv) when the orbital forcing of the global glaciation-climate model is accompanied by the alkalinity ($CO_2$) forcing containing a dominant-period shift from 41 kyr to 80 kyr, this ice-climate system produces a 40-to-100 kyr glacial rhythmicity transition resembling the MPT LR04 data, and (v) when the glaciation-climate model is forced by an alkalinity ($CO_2$) forcing containing a periodicity transition from 20 kyr to 42 kyr, a non-linear interplay of the orbital forcing and of ~40-kyr periods of the alkalinity forcing may produce glaciation periods of ~100 kyr that are also consistent with the LR04 data.

## 1. Introduction

Around 1 Myr ago, the dominant period of the glacial-interglacial cycles shifted from ~41 to ~100 kyr. The disambiguation of this change in glacial rhythmicity, i.e., the Middle Pleistocene Transition, or MPT hereafter, has been a challenge for the scientific community throughout the last few decades (e.g., Saltzman and Verbitsky, 1993; Clark and Pollard, 1998, Tziperman et al., 2006; Peacock et al., 2006; Abe-Ouchi et al., 2013; Crucifix, 2013; Mitsui and Aihara, 2014; Paillard, 2015; Ashwin and Ditlevsen, 2015; Verbitsky et al., 2018; Willeit et al., 2019; Riechers et al., 2022; Shackleton et al, 2023; Carrillo et al, 2025; Scherrenberg et al., 2025; Pérez-Montero et al., 2025). Since the orbital insolation forcing does not change its pattern through the MPT, several proposed hypotheses included slow changes in governing parameters *internal to the Earth System*. These may define intensities of positive (e.g., variations in carbon dioxide concentration, Saltzman and Verbitsky, 1993) or negative (e.g., regolith erosion, Clark and Pollard, 1998) system feedbacks or a combination of positive and negative feedbacks (e.g., the interplay of ice-sheet vertical temperature advection and the geothermal heat flux, Verbitsky and Crucifix, 2021). The importance of the orbital forcing in generating the pre-MPT ~41 kyr cycles and post-MPT ~100 kyr cycles has widely been acknowledged. In particular, it has been suggested that orbital periods either directly drive these cycles (Raymo et al., 2006; Bintanja and Van de Wal, 2008; Tzedakis et al., 2017, Barker et al, 2025) or synchronize auto-oscillations of the Earth's climate (Saltzman and Verbitsky, 1993, Tziperman et al., 2006, Rial et al., 2013; Nyman and Ditlevsen, 2019; Shackleton et al., 2023). However, the orbital forcing has not been considered to play a role in the origin of the MPT.

Recently, it has been proposed (Ma et al., 2024) that the amplitude of the orbital forcing may experience a change on a million-year timescale and this may have its effect on the MPT. Verbitsky and Volobuev (2025) suggested that the orbital forcing may play an even bigger role and can also change the dynamical properties of the Earth's climate system. For example, it may change the timescale of the

vertical advection of mass and temperature in ice sheets and make their dynamics sensitive to initial
values. Is ice physics unique in this sense? To answer this question, in this paper we will consider the
calcifier-alkalinity (C-A) model that describes entirely different physics, focusing on the interactions
between a population of calcifying organisms and ocean alkalinity (Omta et al., 2013). Previously, it has
been shown that:
(a) The C-A system relaxes slowly to its asymptotic state, i.e., it has a long memory of its initial
conditions (Omta et al., 2013);
(b) The asymptotic state of the orbitally forced C-A system depends on its initial conditions (Omta et
al., 2016).
We will demonstrate here that the relaxation of the dominant period of the orbitally forced C-A
system from its initial value to the asymptotic value can include a sharp transition similar to the MPT. We
will also perform a scaling analysis of the C-A model and demonstrate that the asymptotic dominant
periods are defined by a conglomerate similarity parameter combining the amplitude of the orbital forcing
and the initial values. In other words, *the orbital forcing enables the dominant-period sensitivity to initial*
*values.* We will also prove that what we call an MPT-like event in terms of the alkalinity periodicity can
be translated into an MPT event in terms of the glacial rhythmicity.
## 2. Ocean calcifier-alkalinity model
The C-A model was first formulated by Omta et al. (2013) and focuses on the throughput of alkalinity
through the World's oceans. The alkalinity is a measure for the buffering capacity of seawater that
controls its capacity for carbon storage through the carbonate equilibrium (Broecker and Peng, 1982;
Zeebe and Wolf-Gladrow, 2001; Williams and Follows, 2011). Alkalinity is continuously transported into
the oceans as a consequence of rock weathering on the continents. When alkalinity is added to the ocean,
the solubility of $CO_2$ increases leading to an uptake of carbon from the atmosphere into the ocean.
Removal of alkalinity from the water (through incorporation of calcium carbonate into the shells of
calcifying organisms and subsequent sedimentation) leads to a lower $CO_2$ solubility and thus outgassing
of carbon from the ocean into the atmosphere. The C-A model assumes that alkalinity $A$ (mM eq) enters
the ocean at a constant rate $I_0$ (mM eq yr$^{-1}$). Alkalinity is taken up by a population of calcifying
organisms $C$ (mM eq) growing with rate constant $k$ ((mM eq)$^{-1}$ yr$^{-1}$) and sedimenting out at rate $M$ (yr$^{-1}$).
Altogether, the model equations are:
$$\frac{dA}{dt} = I_0 - kAC \tag{1}$$
$$\frac{dC}{dt} = kAC - MC \tag{2}$$
with *t* the time (yr). Since there exists observational evidence of variations in calcifier productivity
correlated with Milankovitch cycles (Beaufort et al., 1997; Herbert, 1997), we include a periodic forcing
term in the calcifier growth parameter $k$:
$$k = k_0 \left(1 + \alpha cos\left(\frac{2\pi t}{T}\right)\right) \tag{3}$$
As in Omta et al. (2016) and Shackleton et al. (2023), $k_0$ is the average value of $k$, $\alpha$ is the non-
dimensional forcing amplitude, and $T$ (yr) is the forcing period.
Generally speaking, the alkalinity budget is also affected by the seawater carbonate saturation state.
In particular, calcite preservation tends to increase with increasing carbonate ion concentration (Broecker
and Peng, 1982; Archer, 1996). This carbonate compensation feedback was included in the detailed multi-
box version of the calcifier-alkalinity model (Omta et al., 2013). Essentially, carbonate compensation
acted as a negative feedback that enhanced the damping of the cycles. If the periodic forcing was
sufficiently strong to overcome this damping, then the model behavior was very similar to the behavior of
the model without carbonate compensation (see Fig. 5 in Omta et al., 2013). Here we chose to use the
simpler, more parsimonious model.
Simulations with the C-A model are performed in Julia version 1.11.2. As in Shackleton et al. (2023),
we use the KenCarp58 solver (Rackauckas and Nie, 2017) with a tolerance of $10^{-16}$ (code is available on
GitHub – https://github.com/AWO-code/VerbitskyOmta).
**3. Results and Discussion**
The C-A system (1) – (3) produces sawtooth-shaped cycles in alkalinity, with the alkalinity rising
slowly and declining steeply. This corresponds to $CO_2$ decreasing slowly and increasing rapidly,
consistent with the ice-core record (Lüthi et al., 2008). In Fig. 1, a simulation with initial conditions
$A(0) = 2.0$ mM eq, $C(0) = 4 * 10^{-5}$ mM eq, forcing strength $\alpha = 0.012$, forcing period $T = 40$ kyr (the
obliquity period, i.e., year-average insolation), and reference values for other parameters (Omta et al.,
2016) is shown.

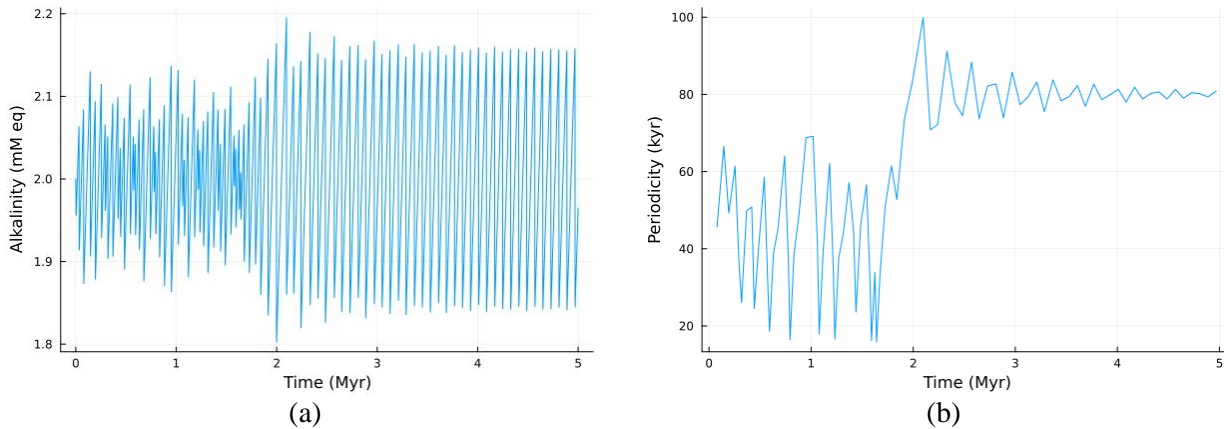

(a)                                           (b)

**Figure 1.** C-A system under orbital forcing ($A(0) = 2.0$ mM eq, $C(0) = 4 * 10^{-5}$ mM eq, $\alpha = 0.012$, $T =$
40 kyr): (a) alkalinity, (b) dominant period as a function of time
The dominant period initially evolves around the forcing period of 40 kyr, then sharply (MPT-
like) increases to about 80 kyr (twice the forcing period) and stabilizes at this level. This period shift
occurs through a different mechanism than in earlier studies using the C-A model, where period shifts
involved noise (Omta et al., 2016) or a positive feedback (Shackleton et al., 2023) to "kick" the system
from one dominant period to another one. Here no such kick is imposed: the period shift rather emerges
as part of the transient dynamics of the system, as it relaxes from its initial towards its asymptotic state.
For the first ~1.7 Myr of the simulation, there appears to be an approximate but not exact frequency lock,
from which the system has difficulty escaping. Once the system is out of this approximate frequency lock,
its period increases relatively rapidly until it reaches another multiple of the forcing period where the
system becomes locked again.
In the following, we analyze how the ***initial*** and ***asymptotic*** periods may depend on the system
parameters. In particular, we formulate a scaling law (Section 3.1) that we then investigate in more details
through simulations (Section 3.2). In Section 3.3 we project the discovered alkalinity dynamics onto the
glacial rhythmicity.

**3.1 Scaling law**
137         The C-A system of equations (1) – (3) contains seven governing parameters, including the initial
conditions. Both the mean initial and the asymptotic periods have to be functions of these seven
parameters. Thus, we can write:
$P = \varphi(I_0, k_0, \alpha, T, M, A(0), C(0))$ (4)
with $P$ the asymptotic period. If we take $I_0, k_0$ as parameters with independent dimensions, then
according to the $\pi$-theorem (Buckingham, 1914):

$$\frac{P}{\tau} = \Phi\left[\alpha, \frac{T}{\tau}, M\tau, \frac{A(0)}{F}, \frac{C(0)}{F}\right]$$ (5)

Here $\tau = (k_0 I_0)^{-1/2}, F = \left(\frac{I_0}{k_0}\right)^{1/2}$.
In this study, we will focus just on two similarity parameters $\alpha, \frac{A(0)}{F}$ leaving $\frac{T}{\tau}, M\tau, \frac{C(0)}{F}$ to remain
constant:

$$\frac{P}{\tau} = \Phi\left[\alpha, \frac{A(0)}{F}\right]$$ (6)
Using similar reasoning, we can write for the initial period $P_0$:

$$\frac{P_0}{\tau} = \Psi\left[\alpha, \frac{A(0)}{F}\right]$$ (7)
**3.2 Scaling law simulations**
To investigate the scaling laws (6, 7), we perform a suite of 10-Myr simulations in which we vary $\alpha$ and
$\frac{A(0)}{F}$. The average periods during the first 1 Myr ($P_0$) and the last 1 Myr ($P$) as a function of $\alpha$ and $\frac{A(0)}{F}$
are presented in Figs. 2a and 2b, respectively. The range in $A(0)$, which determines the vertical axis range
in Fig. 2, was chosen based on the estimated total weathering input of $CaCO_3$ (Milliman et al., 1999),
which could give rise to alkalinity variations of up to ~20% on ~100-kyr timescales (Omta et al., 2013).
The lower and higher ends of the range are probably a bit less likely than the middle part of the range.
There is no obvious constraint on α (horizontal axis in Fig. 2), which is why we varied that parameter by
two orders of magnitude. In total, Fig. 2 encompasses the results of 12,798 simulations.

(a)

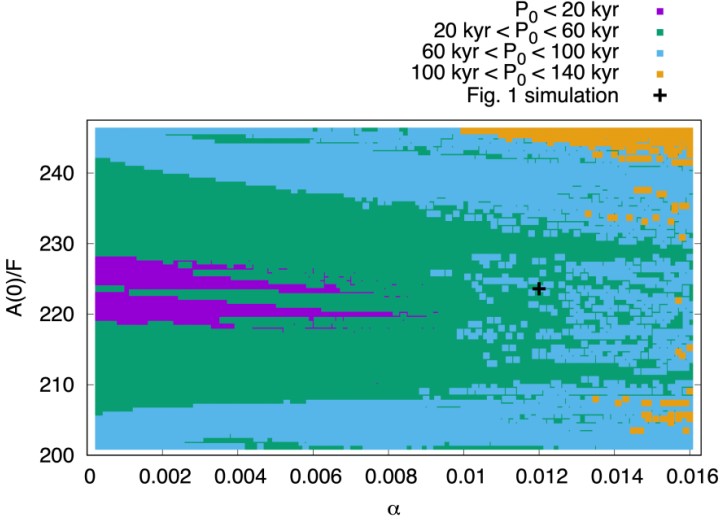

(b)

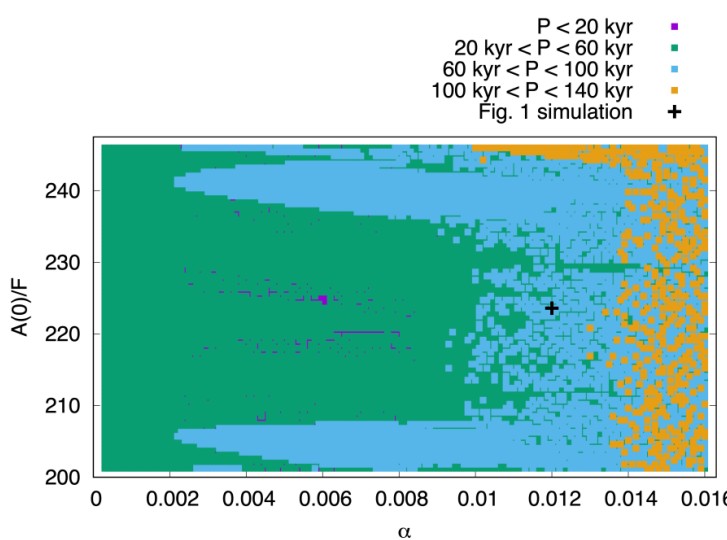

**Figure 2.** (a) Initial periods $P_0$ (average of first 1 Myr of 10-Myr simulations), and (b) asymptotic periods $P$ (average of last 1 Myr of 10-Myr simulations). Each dot represents one simulation; the black cross indicates the parameter values for the simulation shown in Fig 1. Two similarity parameters are varied: the non-dimensional forcing strength $\alpha$ (horizontal axis) and the scaled initial alkalinity $\frac{A(0)}{F}$ (vertical axis). In total, Fig. 2 encompasses the results of 12,798 simulations. In all simulations, $T = 40$ kyr and $C(0) = 4 * 10^{-5}$ mM eq. Other parameters are kept constant at their reference values (Omta et al., 2016):

$M = 0.1$ yr$^{-1}$, $k_0 = 0.05$ (mol eq)$^{-1}$ m$^3$ yr$^{-1}$, $I_0 = 4 * 10^{-6}$ mol eq m$^{-3}$ yr$^{-1}$, i.e., $F = \left(\frac{I_0}{k_0}\right)^{\frac{1}{2}} = 0.00894$.

178
179 From Fig. 2, it can be observed that:

(a) $P_0$ and $P$ depend on $\alpha$ and $\frac{A(0)}{F}$ in different manners. Most obviously, $P_0 < 20$ kyr in a significant fraction of the simulations whereas $P > 20$ kyr in almost every simulation. Furthermore, $P > 100$ kyr occurs in many more simulations than $P_0 > 100$ kyr. These differences imply that a period shift emerges in a significant fraction of the simulations.

(b) When $\alpha \to 0$, the asymptotic period $P$ becomes independent of the initial value $A(0)$ (Fig. 2b), which means that the similarity parameters $\alpha, \frac{A(0)}{F}$ in the C-A system (1) – (3) collide into one conglomerate similarity parameter $\alpha^x \left[\frac{A(0)}{F}\right]^y$ (the parameters $x$ and $y$ should be determined experimentally). This then provides us with the final form of the scaling law for the asymptotic period:

$$\frac{P}{\tau} = \Phi\left\{\alpha^x \left[\frac{A(0)}{F}\right]^y\right\} \tag{8}$$

The scaling law (8) implies that the *orbital forcing affects the dynamical properties of the C-A physics enabling the sensitivity of asymptotic periods to initial values*.

(c) When $\alpha$ increases (e.g., Gough, 1981, Ma et al., 2024), the sensitivity of the dominant asymptotic period to the initial conditions $\frac{d\left(\frac{P}{\tau}\right)}{d\left(\frac{A(0)}{F}\right)}$ also increases. Specifically, when $\alpha < 0.002$, as we have already noted, $\frac{P}{\tau}$ is not sensitive to initial values. When $0.002 < \alpha < 0.01$, it takes $\Delta\left(\frac{A(0)}{F}\right) \sim 10$ to obtain a different asymptotic period. Orbital forcing with $0.01 < \alpha < 0.014$ reduces the critical value of initial values changes to $\Delta\left(\frac{A(0)}{F}\right) \sim 1$, and finally for $\alpha > 0.014$ changes as small as $\Delta\left(\frac{A(0)}{F}\right) \sim 0.1$ lead to different asymptotic periods.

(d) Depending on $\alpha^x \left[\frac{A(0)}{F}\right]^y$, the periodicity transition could have been not just of the $40 - 80$ kyr type (as shown in Fig. 1), but also of a $20 - 40$, $40 - 120$, or even $80 - 40$ kyr type (Fig. 3).

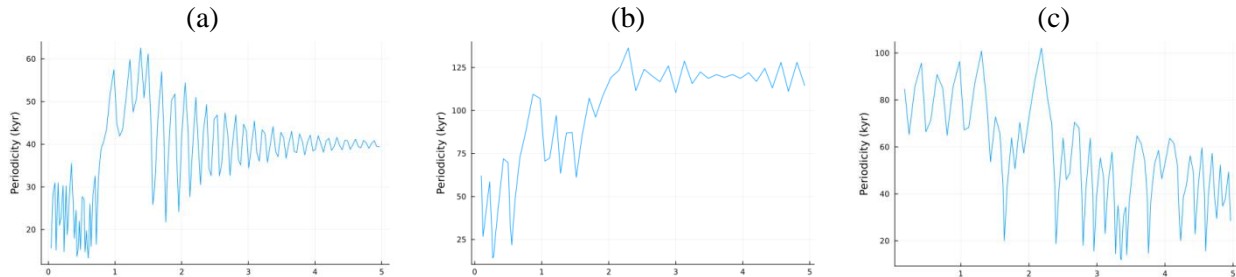

**Figure 3.** Alkalinity dominant-period transitions of (a) $20 - 40$ kyr ($A(0) = 2.0$ mM eq, $\alpha = 0.008$); (b) $40 - 120$ kyr ($A(0) = 2.0$ mM eq, $\alpha = 0.0149$); and (c) $80 - 40$ kyr ($A(0) = 1.82$ mM eq, $\alpha = 0.0122$). In all experiments $C(0) = 4 * 10^{-5}$ mM eq, $T = 40$ kyr.

Most of the simulations reach their asymptotic periods within the first 1 Myr. A period shift after 1 Myr occurs in 3,217 out of the 12,798 simulations (about 25%) represented in Fig. 2. However, it is impossible to infer from the proxy data how common or rare a shift in the dominant period of the glacial-interglacial cycles actually is in the real World, since the observed Pleistocene climate is essentially a single time series.

Classical phase locking (e.g., Tziperman et al., 2006) requires some kind of dissipation in the dynamical system that erases the memory of its initial values. Obviously, this is not the case with the dominant-period trajectories we observe in Figs. 1, 2, and 3. At the same time, the asymptotic periods are multiples of the forcing period. We therefore suggest calling this phenomenon a *delayed* phase locking.

**3.3 Translating alkalinity dynamics into glacial rhythmicity**

To investigate the link between the modelled relaxation process and the climate system, we applied some alkalinity time series to the Verbitsky et al. (2018) model as additional forcings for the ice mass balance. This model has been derived from the scaled mass-, momentum-, and heat-conservation equations of non-Newtonian ice flow combined with an energy-balance model of global climate. In our experiments, all reference parameters of the Verbitsky et al. (2018) model remain the same, except one parameter that affects the intensity of positive feedbacks. On its own accord, the Verbitsky et al. (2018) model can produce a period shift if a positive feedback is sufficiently strong. We now set this positive feedback weaker to deprive the Verbitsky et al. (2018) model of this ability to produce MPT-like events.

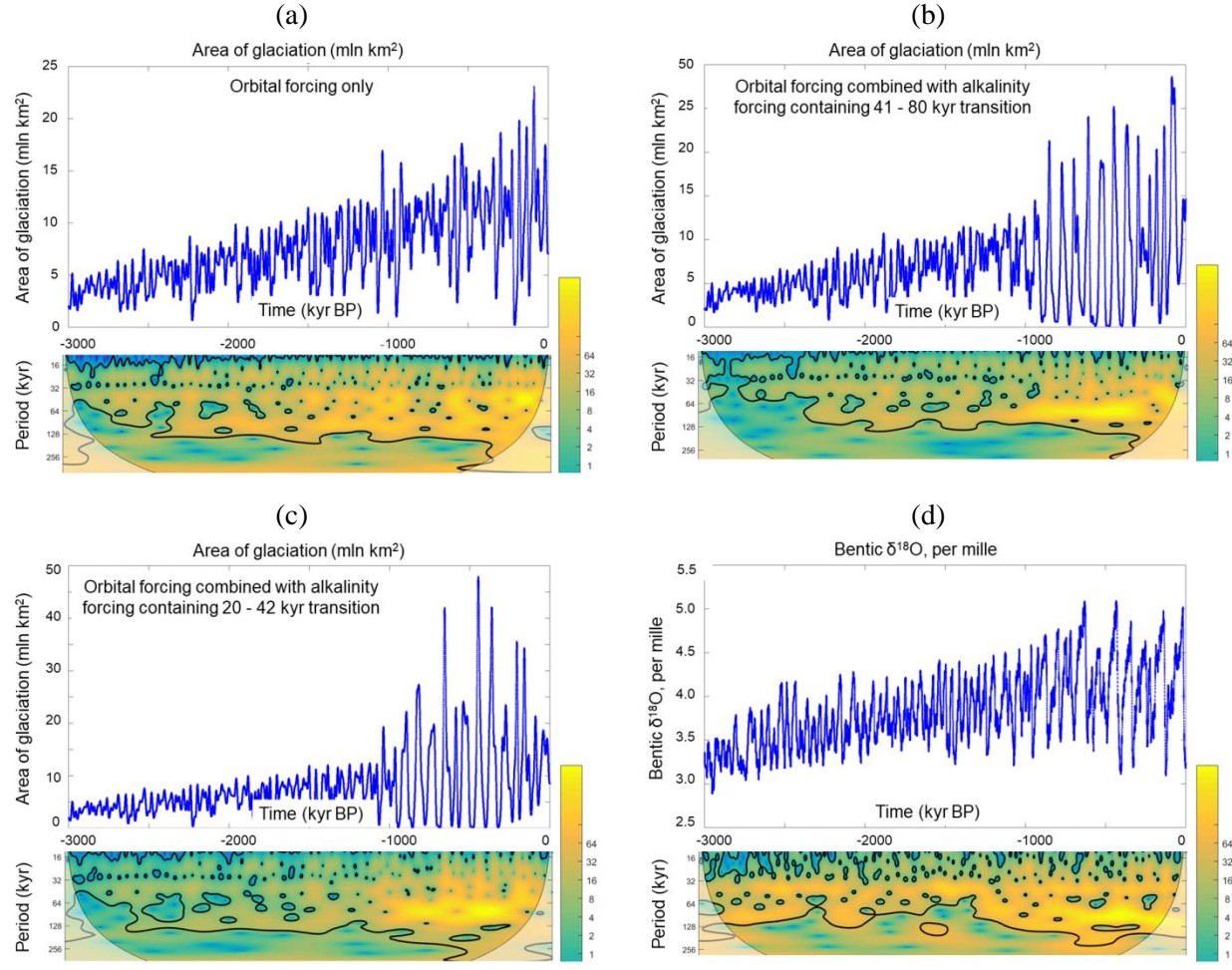

**Figure 4.** Ice–climate system (Verbitsky et al., 2018) response to a pure orbital (a) and to a combination of orbital and alkalinity ($CO_2$) forcing (b - additional alkalinity ($CO_2$) forcing contains a periodicity transition from 41 kyr to 80 kyr, c - additional alkalinity ($CO_2$) forcing contains a periodicity transition from 20 kyr to 42 kyr) presented as time series and evolutions of wavelet spectra over 3 Myr for calculated ice-sheet glaciation area ($10^6$ km$^2$) (a, b, c) and for the Lisiecki and Raymo (2005) benthic $\delta^{18}$O record (d). The vertical axis of wavelet spectra is the period (kyr); the horizontal axis is time (kyr before present).The color scale shows the continuous Morlet wavelet amplitude, the thick line indicates the peaks with 95 % confidence, and the shaded area indicates the cone of influence for the wavelet transform.

In Figure 4a, we show the weak-positive-feedback area-of-glaciation evolution under the imposed
cooling trend without additional alkalinity ($CO_2$) forcing. This time series does not exhibit MPT-like
periodicity changes. When an additional alkalinity ($CO_2$) forcing containing a period shift from 41 kyr to
80 kyr is applied, the glaciation-climate system produces a 40-to-100 kyr glacial rhythmicity transition
resembling the LR04 data (Figure 4b *vs* 4d). This is the case of the direct alkalinity-forced period
transition that could probably be anticipated. Yet, it is quite remarkable and very unintuitive that the
alkalinity forcing may entertain a more subtle interplay with the direct orbital forcing. This becomes
evident in the experiment when we forced the Verbitsky et al. (2018) model with an alkalinity ($CO_2$)
forcing containing periodicity transitions from 20 kyr to 42 kyr. A non-linear interplay of the direct
orbital forcing (i.e., mid-July insolation at $65^0$N, Berger and Loutre, 1991) and of ~40-kyr periods of the
alkalinity forcing may produce glaciation periods of ~100 kyr also consistent with the LR04 data (Figure
4c *vs* 4d).
In this paper, we do not aspire to precisely reproduce the empirical time series and by doing so to
claim any specific attribution. However, with the above experiments, we demonstrate that the calcifier-
alkalinity dynamics may have a profound effect on the climate system, and what we call an MPT-like
event in terms of the alkalinity periods can be translated into an MPT event in terms of glacial
rhythmicity.
Generally speaking, it would be indeed interesting to explore possible interactions of different
initial-value-sensitive systems such as the glaciation-climate system (Verbitsky and Volobuev, 2025), the
calcifier-alkalinity system (this presentation), or, possibly, carbon cycle system (Carrillo et al., 2025).
Since the MPT was a global, almost synchronous, event, the discovering of the synchronization
mechanism may be an important next step. In our experiments, presented in Fig. 4, the alkalinity ($CO_2$),
together with the direct orbital forcing, acted as the external synchronizing force for the glaciation-climate
system, but many other scenarios are, indeed, possible.
**4. Conclusions**
The history of climate has been given to us as a single time series. For many years, perhaps
somewhat naively, significant efforts have been applied to reproduce this time-series under a unique
combination of the governing parameters and thus presumably to explain the history. The fundamental
fact that the dominant-period trajectory is governed by a conglomerate similarity parameter
$\alpha^x \left[\frac{A(0)}{F}\right]^y$ (demonstrating a property of incomplete similarity as defined by Barenblatt, 2003) tells us that
the MPT could have been produced under very different combinations of the intensity of orbital forcing
and initial values. Furthermore, the scaling laws (7) and (8), as they are presented in Fig. 2, show that not
only periodicity transitions of the 40 – 80 kyr type (as we observe in the available data), but also of 20 –
40, 40 – 120, or even 80 – 40 kyr types would be possible. Some of these transitions, i.e., 40 – 80, 40 –
120, and, remarkably, 20 – 40 kyr types, produce glaciation MPT events consistently with the data. Most
intriguingly, the conglomerate similarity parameter $\alpha^x \left[\frac{A(0)}{F}\right]^y$ implies that such an "intimate" terrestrial
property as the sensitivity of alkalinity-calcination system to initial values manifests itself only under
orbital forcing, and thus *the MPT exhibits a remarkable physical phenomenon of orbitally enabled*
*sensitivity to initial values.*
We focused our paper on the past, MPT, event. Nevertheless, since we force our model with a
generic obliquity, 40-kyr, forcing without a particular connection to the celestial time, any time series out
of 12,798 simulations can be assumed as starting at present, and any observed periodicity transition can
be considered not just as possible past transition but as potential future transition as well. Therefore, the
complexity of Fig. 2 demonstrates not just the empirical data disambiguation challenge, but also the
difficulty of the future climate prediction.
In this paper, when we establish a consistency between model results and empirical data, we are
talking about periodicity transitions only, leaving purposely amplitude-periodicity relationship outside the
scope. Verbitsky and Crucifix (2020) demonstrated that in the short-memory ice-climate system,
independent on initial values, the relationship between the glacial area amplitude $S'$ and duration of
glacial cycles $P$ is governed by a property of scale invariance, such that $S' \sim P^2$. For the calcifier-
alkalinity system, which has a long memory and dependence on the initial values, there exists a linear
relationship between the amplitude and the period of the cycles. As was explained in Omta et al. (2016),
this property emerges because a longer duration of the slow linear increase in alkalinity (determined by
the constant term $I_0$ in equation (1)) implies proportionately larger amplitude of the cycle and vice versa.
In the future, it will be highly desirable to compare predictions from different models regarding the
periodicity and amplitude of variations in ice volume and global mean surface temperature against the
newly available sea-level and global mean surface temperature data (Clark et al, 2025).
**Competing interests:** The authors declare that they have no conflict of interest.
**Code availability.** Simulations with the C-A model are performed in Julia version 1.11.2 using the
KenCarp58 solver (Rackauckas and Nie, 2017) with a tolerance of $10^{-16}$ (https://github.com/AWO-
code/VerbitskyOmta).
**Data availability.** This paper refers exclusively to published research articles and their data. We refer the
reader to the cited literature for access to the data.
**Author contributions:** MYV conceived the research, AWO performed the simulations and discovered
the MPT-like periodicity relaxation, MYV performed the scaling analysis and discovered the orbitally
enabled sensitivity to initial values. The authors jointly wrote and edited the paper.
**Acknowledgements:** We are grateful to our three anonymous reviewers for their insightful reviews.

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
