# Peer review of "Rapid Communication: Middle Pleistocene Transition as a Phenomenon of"

_EGUsphere, 2025_

## Author Comment (AC1)

Dear Anonymous Reviewer,

Thank you for reviewing our manuscript and providing insightful feedback. Below, we reply to your comments (marked as **bold blue**) and propose several changes to the manuscript motivated by your suggestions.

**The Rapid Communication manuscript by Verbitsky and Omta describes the relaxation behavior when an idealized model of the ocean's alkalinity budget is subjected to idealized orbital forcing, documenting spontaneous changes in the dominant periodicity of the model response. The manuscript draws an interesting comparison to the Mid-Pleistocene Transition from obliquity-pacing of climate to a saw-tooth pattern with ~100kyr dominant periodicity, but it offers little discussion why the dynamic behavior of the idealized model should apply to the real Earth System. Because of the abbreviated format of the manuscript it is difficult to assess the significance of the work.**

**Response:** Your general comment consists of two parts and we would like to respond to them separately.

First, we definitely appreciate that you consider our findings to be interesting. We would like to clarify though that the essence of them is not simply a spontaneous change of the dominant periodicity, but a strong dependence of this process on initial values. And finally – the most intriguing part of this phenomenon - this dependence on initial conditions is enabled by the orbital forcing. When the orbital forcing is weak, the asymptotic period is initial-values independent. A strong orbital forcing makes these periods highly sensitive to initial values. We tried to underline this by bringing this observation to the title of the paper.

We understand the second part of your comment ("**why the dynamic behavior of the idealized model should apply to the real Earth System**") as your concern about the physical content of the model. Such concern is very appropriate. Over the last two decades, the field of Pleistocene glacial-rhythmicity studies has been overwhelmed by research based on so called "conceptual" or phenomenological models that do not have any physical basis except their ability (often artificially forced by Boolean statements) to reproduce the empirical record. Verbitsky and Crucifix (2023) have warned the scientific community that conceptual models may simply not have a physical similarity with nature and therefore add little to our understanding of it. Accordingly, we selected a model that is based on the physically explicit ocean alkalinity budget. Hence, this is certainly a development in the direction you so rightly advocate for.

Let us offer you a big picture that (because of the "**abbreviated format**") may not have been articulated extensively enough. It is not our intention to claim that the discovered phenomenon is a single possible explanation of the Mid-Pleistocene Transition (MPT). In fact, one of the authors expended significant efforts to demonstrate that because of the fundamental properties of viscous ice mass- and heat-conservation equations, the MPT could be an outcome of multiple scenarios of completely different nature (Verbitsky, 2022). Moreover, it has recently been discovered (Verbitsky and Volobuev, 2024) that the orbital forcing may enable sensitivity of the ice-climate system to initial values, which provides even more MPT scenarios.

We started our experiments with the ocean alkalinity-calcification system because we wanted to see how general this phenomenon (orbitally enabled sensitivity to initial values) is and indeed, we found it in this system as well. It would be relatively easy now to write the

mass-balance equation of the ice sheet with the alkalinity (or $CO_2$) as the forcing on the right-hand side of it and to reproduce the empirical record under "reasonable assumptions" about unconstrained parameters. However, this is exactly what we do not want to do, because it would be yet another fitting exercise that does not prove a scenario is unique but simply demonstrates that it is within the range of admissible parameters. Instead, we want the scientific community to realize that *a single empirical time series that is given to us by nature is in fact very fragile and it could have been very different under subtle changes of the million-years-old initial values of ocean alkalinity*. It is not, indeed, the Saltzman-Lorentz "butterfly" effect but it is reminiscent of it (See also Fig. AC1-1 below).

**Action:** We will articulate more clearly both the essence of our observation and the goal of our study.

**Detailed comments:**

**1. Orbital forcing of the calcification rate constant as the primary driver of CO2 change is a highly unusual model to use, and simulating the ocean's alkalinity budget completely independent of seawater carbonate saturation state is questionable. This model may be suitable if the point of the manuscript is simply to document "a remarkable physical phenomenon", but drawing any conclusions about the paleoclimate record based on these results would require detailed justification of the model and discussion of its applicability.**

**Response:** Yes, the point of the manuscript is simply to document a remarkable physical phenomenon and we are glad that you find the model to be suitable for this purpose.

Having said this, we agree that the ocean's alkalinity budget is affected by the seawater carbonate saturation state. In particular, calcite preservation tends to increase with increasing carbonate ion concentration (Broecker and Peng, 1982; Archer, 1996). This carbonate compensation feedback was included in the detailed multi-box version of the calcifier-alkalinity model (Omta et al., 2013). Essentially, carbonate compensation acted as a negative feedback that enhanced the damping of the cycles. If the periodic forcing was sufficiently strong to overcome this damping, then the model behavior was very similar to the behavior of the model without carbonate compensation (see Fig. 5 in Omta et al., 2013). Here we chose to use the simpler, more parsimonious model.

As Grigory Barenblatt (2003) said, "applied mathematics is the *art* of constructing mathematical models of phenomena in nature". It is an art because there are no strict rules about model design, and it often takes the intuition of a scientist to select which physics is the cornerstone of the model. We study ice ages and therefore, for many years, the physics of ice flow was a natural choice for building ice-age models for many scientists (including one of the authors). Even so, the ocean alkalinity cycle operates on these same orbital timescales. Orbital forcing of the calcification rate constant may be "unusual", but "unusual" is not a physical argument, and we have to talk about physical feasibility instead. As we have mentioned (lines 85-86), "…there exists observational evidence of variations in calcifier productivity correlated with Milankovitch cycles (Beaufort et al., 1997; Herbert,

1997)"; orbital forcing of the calcification rate constant seems therefore to be a reasonable possibility.

**Action:** We will add above discussion to the revised version of the paper.

**2.The authors draw attention to the fact that the model remains phase locked to the forcing frequency for millions of years before spontaneously settling on oscillation with a dominant period that appears to be an integer multiple of the forcing period. The authors should explain how their finding is similar or different to the notion of skipping obliquity cycles advanced by Wunsch and Huybers. Is this simply a case of non-linear phase locking?**

**Response:** Non-linear phase locking is an initial-values independent process (Tziperman et al., 2006, we are quoting this paper by Tziperman, Raymo, Huybers, and Wunsch because it is more detailed than the earlier papers by Wunsch and Huybers). In our model, the asymptotic periodicity of 40, 80, or 120 kyr depends on initial values and this dependence on initial values is enabled by the orbital forcing. When the orbital forcing is weak, the asymptotic period is initial-values independent. A strong orbital forcing makes these periods highly sensitive to the initial values.

**Action:** We will articulate more clearly the essence of our observation relative to non-linear phase locking.

**3. Given the emphasis on the million-year persistence of influence from the model initial values it is worth noting that the model does not include any stochastic "white noise" term that would over time erode in initial value information. It would have been helpful if Figure 1 included a small set of identically forced simulations with different initial conditions, to assess if they relax onto the same long-term solution. Also, it would have been helpful if the manuscript included power spectra and phase space portraits for the different solution groups indicated in Figure 2b.**

**Response:** It is unfortunate that Fig. 3 somehow escaped your attention. This figure represents exactly what you are asking for, and not for a "small set" but for 12,798 model experiments. All our findings are based on these experiments.

Though we will not be able to show all 12,798 time series, having some samples is certainly a good idea. Since all our scaling laws and results are focused on periodicity, we believe that periodicity time series like Fig. 2b will serve our readers best. In Fig. AC1-1, we show three periodicity time series with slightly different initial conditions A(0)=1.990 (mM eq), A(0)=1.995 (mM eq), and A(0)=2.010 (mM eq). It can be seen that the alkalinity-calcification system has a long memory and the orbital forcing makes it highly sensitive to initial values.

(a)

(b)

(c)

[Figure]

**Figure AC1-1**. C-A system dominant period as a function of time under orbital forcing, $\alpha$ = 0.0134: (a) $A(0) = 1.990$ mM eq, (b) $A(0) = 1.995$ mM eq, (c) $A(0) = 2.010$ mM eq.

**Action:** We will discuss Fig. AC1-1 in the revised version of the paper.

**4. The conclusion takes a major leap from the identified behavior of the model to claiming that "thus MPT exhibits a remarkable physical phenomenon" [line 188]. In absence of any significant discussion on the applicability of the model to the MPT this leap seems rather speculative. Further, it would have been helpful if the manuscript had elaborated on the implications for the interpretation of the dynamic mechanism yielding obliquity-paced iNHG and presumably preconditioning the system to experience some type of MPT. For example, if the model dynamical behavior is applicable then climate change should always lag CO2 change, which always lags orbital forcing by thousands of years.**

**Response:** Let us read lines 185-189 again: "Most intriguingly, the conglomerate similarity parameter also tells us that such an "intimate" terrestrial property as the sensitivity of alkalinity-calcination system to initial values manifests itself only under orbital forcing and thus MPT exhibits a remarkable physical phenomenon of orbitally enabled sensitivity to initial values". Since we are talking here about the alkalinity-calcification system, the statement seems very accurate. Maybe to avoid the impression of a leap, instead of MPT we should call this phenomenon MPT-like events, MPT-type, MPT-resembling events, or so.

With your further suggestion to elaborate **"on the implications for the interpretation of the dynamic mechanism yielding obliquity-paced iNHG"** you seem to try to fit our study into the existing paradigm of the obliquity-paced fluctuations. The whole point of our study is to challenge this paradigm. Specifically, we demonstrate that the terrestrial climate system has a long memory; the orbital forcing makes the ocean chemistry highly sensitive to initial values, and altogether it may make Earth climate highly unpredictable (see Fig. AC1-1). Furthermore, causal relationships between variables do not necessarily align with temporal leads and lags in complex nonlinear systems such as the climate (e.g., Van Nes et al., 2015, Verbitsky et al 2019).

**Action:** We will add the above discussion into the revised version of the paper.

**References**

Archer, D. E.: An atlas of the distribution of calcium carbonate in sediments of the deep sea, Global Biogeochem. Cycles, 10, 159–174, 1996.

Beaufort, L., Lancelot, Y., Camberlin, P., Cayre, O., Vincent, E., Bassinot, F., and Labeyrie, L.: Insolation cycles as a major control of Equatorial Indian Ocean primary production, Science, 278, 1451–1454, 1997.

Barenblatt, G. I.: Scaling, Cambridge University Press, Cambridge, ISBN 0 521 53394 5, 2003.

Broecker, W. S., and Peng, T. H.: Tracers in the Sea, Lamont-Doherty Geological Observatory, Palisades, NY, 1982.

Herbert, T.: A long marine history of carbon cycle modulation by orbital-climatic changes, Proc. Natl. Acad. Sci., 94, 8362–8369, 1997.

Omta, A. W., Van Voorn, G. A. K., Rickaby, R. E. M., and Follows, M. J.: On the potential role of marine calcifiers in glacial-interglacial dynamics, Global Biogeochem. Cycles, 27, 692–704, 2013.

Tziperman, E., Raymo, M. E., Huybers, P., and Wunsch, C.: Consequences of pacing the Pleistocene 100 kyr ice ages by nonlinear phase locking to Milankovitch forcing, Paleoceanography, 21, PA4206, https://doi.org/10.1029/2005PA001241, 2006.

Van Nes, E. H., Scheffer, M., Brovkin, V., Lenton, T. M., Ye, H., Deyle E., Sugihara, G.: Causal feedbacks in climate change, Nat. Clim. Change, 5, 445–448, 2015.

Verbitsky, M. Y.: Inarticulate past: similarity properties of the ice–climate system and their implications for paleo-record attribution, Earth Syst. Dynam., 13, 879–884, https://doi.org/10.5194/esd-13-879-2022, 2022.

Verbitsky, M. Y. and Crucifix, M.: Do phenomenological dynamical paleoclimate models have physical similarity with Nature? Seemingly, not all of them do, Clim. Past, 19, 1793–1803, https://doi.org/10.5194/cp-19-1793-2023, 2023.

Verbitsky, M. and Volobuev, D.: Milankovitch Theory "as an Initial Value Problem", EGUsphere [preprint], https://doi.org/10.5194/egusphere-2024-1255, 2024.

Verbitsky, M. Y., Mann, M. E., Steinman, B. A., and Volobuev, D. M.: Detecting causality signal in instrumental measurements and climate model simulations: global warming case study, Geosci. Model Dev., 12, 4053–4060, https://doi.org/10.5194/gmd-12-4053-2019, 2019.

---

## Author Comment (AC2)

Dear Anonymous Reviewer,

Thank you for reviewing our manuscript and providing insightful feedback. Below, we reply to your comments (marked as **bold blue**) and propose several changes to the manuscript motivated by your suggestions.

**Summary:**

**The Rapid Communication article by Verbitsky and Omta presents a sensitivity study carried out on a simple conceptual ocean chemistry model. The underlying model used in this study is the calcifier-alkalinity (CA) model, as described by Omta et al. (2013, doi: 10.1002/gbc.20060). The authors apply an obliquity-paced orbital forcing to the calcifier growth parameter and demonstrate that the system exhibits long equilibrium times, with transitions from an initial dominant period to an asymptotic one that can occur abruptly. The authors find that this transition in the period can be highly sensitive to the initial conditions and that this sensitivity depends on the amplitude of the orbital forcing. Based on their results, the authors state that the MPT could be the result of a relaxation process resulting in a sharp transition in the dominant period and that it could have resulted from different sets of initial values and orbital forcing amplitudes. Therefore, the observed 41-kyr to 100-kyr shift in the periodicity just resembles one specific instance, but different initial values or an altered orbital forcing amplitude could have led to a completely different pattern in this period shift.**

**In my view, this article presents an interesting view on the MPT. Especially demonstrating that the C-A model can produce abrupt, MPT-like jumps in periodicity purely driven by orbital pacing alone, without the need for any change in parameters, is significant. Furthermore, the result that the sensitivity of the asymptotic state depends on the amplitude of the orbital forcing is very interesting. My primary concern with this work lies in the very conceptual view of the model, and how the results relate to the MPT and the real world. Strengthening the link between the modelled relaxation processes and the real climate system would enhance the significance of the results for the MPT.**

**Response**

We are grateful that you consider our results to be significant and very interesting.

Your suggestion to strengthen the link between the modelled relaxation process and the real climate system has steered us to one more novel result. Specifically, as a step in this direction, we applied some of the modeled alkalinity time series, containing periodicity transitions, as additional forcing to the glacial mass balance of the Verbitsky et al (2018, VCV18 hereafter) model. VCV18 model is a dynamical system, not postulated, but derived from the scaled mass-, momentum-, and heat-conservation equations of non-Newtonian ice flow combined with an energy-balance model of global climate. In our additional experiments, all reference parameters of the VCV18 model remain the same, except one parameter that affects the intensity of positive feedbacks.

[Figure]

**Figure AC2-1.** Ice–climate system response to pure orbital (**a**) and to a combination of orbital and alkalinity (CO$_2$) forcing (**b** - additional alkalinity (CO$_2$) forcing contains periodicity transition from 41 kyr to 80 kyr, **c** - additional alkalinity (CO$_2$) forcing contains periodicity transition from 20 kyr to 42 kyr) presented as time series and evolutions of wavelet spectra over 3 Myr for calculated ice-sheet glaciation area $S$ (10$^6$ km$^2$) (**a**, **b, c**) and for the Lisiecki and Raymo (2005) benthic δ$^{18}$O record (**d**). The vertical axis of wavelet spectra is the period (kyr); the horizontal axis is time (kyr before present).The color scale shows the continuous Morlet wavelet amplitude, the thick line indicates the peaks with 95 % confidence, and the shaded area indicates the cone of influence for wavelet transform.

On its own accord, VCV18 can produce a period shift if a positive feedback is sufficiently strong. We now set this positive feedback weaker to deprive VCV18 of this ability to produce an MPT-like event.

In **Figure AC2-1(a**), we show the weak-positive-feedback VCV18 evolution under the imposed cooling trend without additional alkalinity ($CO_2$) forcing. This time series does not exhibit MPT-like periodicity changes. When the additional alkalinity ($CO_2$) forcing containing periods shift from 41 kyr to 80 kyr is applied, the glaciation-climate system produces a 40-to-100 kyr glacial rhythmicity transition resembling the LR04 data (**Figure AC2-1 (b)** *vs* **(d)**). This is the case of the direct alkalinity-forced period transition that could probably be anticipated. Yet, it is quite remarkable and very unintuitive that the alkalinity forcing may entertain a more subtle interplay with the direct orbital forcing. This becomes evident in the experiment when we forced VCV18 model with an alkalinity ($CO_2$) forcing containing periodicity transitions from 20 kyr to 42 kyr. ***A non-linear interplay of the direct orbital forcing*** (i.e., mid-July insolation at $65^0$N, Berger and Loutre, 1991) ***and of ~40-kyr periods of the alkalinity forcing may produce glaciation periods of ~100 kyr*** also consistent with the LR04 data (**Figure AC2-1 (c)** *vs* **(d)**).

As we have already mentioned in our response to Anonymous Reviewer 1, we do not aspire to precisely reproduce the empirical time series and by doing so to claim any specific attribution. Instead, with the above experiments, we simply demonstrate that the calcifier-alkalinity dynamics may have a profound effect on the climate system, and what we call an MPT event in terms of the period of ***alkalinity*** dynamics can be translated into an MPT event in terms of ***glacial*** rhythmicity.

**Action:** To the extent tolerable by the Rapid Communication format and/or with the Editor's permission, we will include the above discussion in the revised version of the paper.

**Major comments:**

**Please comment on why it is justified to consider T, k0, I0, M and C(0) as constant. For me it is not obvious why P/τ should only depend on α and A(0)/F, but not on Mτ or C(0)/F**

**Response:**

$T, k_0, I_0, M, C(0)$ have been set constant only for the purpose of the current study, as we wanted to focus on the impacts of the orbital forcing and the initial conditions, following, as we mentioned in lines 48-53, the motivation of Verbitsky and Volobuev (2024). Although the orbital period *T* may reasonably be considered constant, we do not have similar constraints on the other parameters.

**Action:** We will clarify this in the revised version of the paper.

**In Fig. 3: is there any physical justification for the used parameter bounds? Can some of the areas in the parameter space be ruled out due to constraints from observations? Based on this analysis, the authors claim that "[...] the MPT could have been not just of the 40 – 80 kyr type, as we observe in the available data, but also of a 20 – 40, 80 – 100, 40 – 120, or even 80 – 40 kyr type" (L.18 f.). Especially the 80-40 kyr scenario, which**

means a reduction in periodicity, seems to occur very rarely in the simulations. It mainly appears in the lower left and upper left parts of Fig 3a) and 3b), where the blue-coloured areas transition to green-coloured areas. Are these scenarios realistic?

**Response:** The range in A(0), which determines the vertical axis range in Fig. 3, was chosen based on the estimated total weathering input of $CaCO_3$ (Milliman et al., 1999), which could give rise to alkalinity variations of up to ~20% on ~100-kyr timescales (Omta et al., 2013). The lower and higher ends of the range, where the 80-to-40 kyr shifts occur, are probably a bit less likely than the middle part of the range. There is no obvious constraint on $\alpha$ (horizontal axis in Fig. 3), which is why we varied that parameter by two orders of magnitude.

**Action:** We will discuss this in the revised version of the manuscript.

Fig. AC1-1 shows that the time until equilibrium is reached is highly variable and for the three shown simulations ranges from ~2 - 6Myr. While this article mainly focuses on the periodicity, the timing until the asymptotic period is reached is important for a full view on the MPT and it would be interesting to include some insights on the mean equilibrium times of the ensemble simulations

**Response:** The main period shift typically occurs within a few Myr from the start of the simulation. Subsequently, there are usually still some minor fluctuations in the period damping out on a timescale of several Myr (see Figs. 2 and AC1-1).

**Action:** We will discuss this in the revised version of the manuscript.

Large parts in Fig.3 do not change in colour. Does this imply that a large quantity of the simulations reach the asymptotic period within the first 1Myr of simulations? Hence, the mentioned shifts in period are only occurring for very specific sets of initial values and forcing amplitudes?

**Response:** Indeed, most of the simulations reach their asymptotic periods within the first 1 Myr. A period shift after 1 Myr occurs in 3,217 out of the 12,798 simulations (about 25%) represented in Fig. 3. In our view, something that happens 25% of the time is not a particularly rare event. Moreover, the observed Pleistocene climate is essentially a single time series. Therefore, it is impossible to infer from the proxy data how common or rare a shift in the dominant period of the glacial-interglacial cycle actually is in the real World.

**Action:** We will discuss this in the revised version of the manuscript.

**Minor comments**

All minor comments have been gratefully accepted and will be taken care of.

**References**

Berger, A. and Loutre, M. F.: Insolation values for the climate of the last 10 million years, Quaternary Sci. Rev., 10, 297–317, 1991.

Lisiecki, L. E. and Raymo, M. E.: A Pliocene-Pleistocene stack of 57 globally distributed benthic δ18O records, Paleoceanography, 20, PA1003, https://doi.org/10.1029/2004PA001071, 2005.

Milliman, J. D., Troy, P. J., Balch, W. M., Adams, A. K., Li, Y. H., and Mackenzie, F. T.: Biologically mediated dissolution of calcium carbonate above the chemical lysocline?, Deep Sea Res. Part I, 46, 1653–1669, 1999.

Omta, A. W., Van Voorn, G. A. K., Rickaby, R. E. M., Follows, M. J.: On the potential role of marine calcifiers in glacial-interglacial dynamics. Glob. Biogeochem. Cycles, 27, 692–704, 2013.

Verbitsky, M. and Volobuev, D.: Milankovitch Theory "as an Initial Value Problem", EGUsphere [preprint], https://doi.org/10.5194/egusphere-2024-1255, 2024.

Verbitsky, M. Y., Crucifix, M., and Volobuev, D. M.: A theory of Pleistocene glacial rhythmicity, Earth Syst. Dynam., 9, 1025–1043, https://doi.org/10.5194/esd-9-1025-2018, 2018.